# Chemo-Diversity and Antiradical Potential of Twelve *Matricaria chamomilla* L. Populations from Iran: Proof of Ecological Effects

**DOI:** 10.3390/molecules24071315

**Published:** 2019-04-03

**Authors:** Elahe Piri, Mohammad Mahmoodi Sourestani, Esmaeil Khaleghi, Javad Mottaghipisheh, Zoltán Péter Zomborszki, Judit Hohmann, Dezső Csupor

**Affiliations:** 1Department of Horticultural Science, Faculty of Agriculture, Shahid Chamran University of Ahvaz, Ahvaz 61357-43311, Iran; elahe.73.piri@gmail.com (E.P.); khaleghi2184@gmail.com (E.K.); 2Department of Pharmacognosy, University of Szeged, Eötvös u. 6, H-6720 Szeged, Hungary; Imanmottaghipisheh@pharmacognosy.hu (J.M.); zombozope@pharmacognosy.hu (Z.P.Z.); hohmann@pharm.u-szeged.hu (J.H.)

**Keywords:** *Matricaria chamomilla* L., Iranian populations, ecological effects, volatile compounds, antiradical capacity

## Abstract

*Matricaria chamomilla* L. is a popular medicinal herb that is used for healing various diseases and is widely distributed worldwide in temperate climate zones, and even in the subtropical climate of Southern and Western Iran. This study was aimed at comparing the volatile oil constituents, along with antiradical potential and HPLC analysis of methanolic extracts from twelve plant samples growing in Iran. The present research was carried out for the first time on these populations. Among seventeen identified volatile chemicals evaluated by GC/MS and GC/FID, representing 92.73–97.71% of the total oils, α-bisabolone oxide A (45.64–65.41%) was the major constituent, except in case of “Sarableh” as a new chemotype, where (*E*)- and (*Z*)-γ-bisabolene (42.76 and 40.08%, respectively) were the predominant components. Oxygenated sesquiterpenes (53.31–74.52%) were the most abundant compounds in the samples excluding “Sarableh” with 91.3% sesquiterpene hydrocarbons. “Sarableh” also exerted the most potent antioxidant capacity with EC_50_ = 7.76 ± 0.3 µg/mL and 6.51 ± 0.63 mmol TE (Trolox^®^ equivalents)/g. In addition, populations “Lali” and “Bagh Malek” contained the highest amounts of apigenin and luteolin with 1.19 ± 0.01 mg/g and 2.20 ± 0.0 mg/g of plant material, respectively. Our findings depict a clear correlation between phytochemical profiles and antiradical potential of *M. chamomilla* and geographical factors.

## 1. Introduction

*Matricaria chamomilla* L. (syn. *Chamomilla recutita* L. Rauschert, German chamomile), belonging to the Asteraceae family, is one of the well-known medicinal plant species which has been widely used for centuries. The herb is currently consumed around the world as herbal tea, with more than 1 million cups daily [1,2]. *M. chamomilla* is used in the treatment of many ailments and disorders; internally to facilitate digestion and as antispasmodic, externally to treat minor wounds. In folk medicine, its use spreads from the relief of various pains such as headaches and toothaches to the facilitation of menstruation [3]. Chamomile essential oil (EO) has been commonly applied in Iranian folk medicine as an anti-inflammatory, antispasmodic, anti-peptic ulcer, antibacterial and antifungal agent [4,5].

Several papers report sesquiterpenes as the most dominant constituent of *M. chamomilla* EO. Spathulenol (12.50%) [6], α-bisabolol oxide A (7.9–62.1%) [7,8,9,10], α-bisabolol oxide B (25.56%) [10], β-farnesene (52.73%) [11], (*E)*-β-farnesene (42.59%) [12], β-cubebene (27.8%) [13], *trans*-β-farnesene [14,15], chamazulene (27.8–31.2%) [16], and α-bisabolol (56.9%) [17] were recently reported as the main EO compounds.

Non-volatile compounds of *M. chamomilla* have also been investigated. The major phytochemicals comprise polyphenols, particularly the flavonoids apigenin, patuletin, quercetin, luteolin and their glucosides [18,19,20]; among them apigenin is reported as one of the most bioactive phenolics [19,21].

In the present study, EO compositions of twelve Iranian *M. chamomilla* populations collected from diverse regions were qualitatively and quantitatively assessed. Since the main radical scavengers are phenolics, flavonoids being the major group within phenolics [22], our study was also designed to determine apigenin and luteolin contents, as its main flavonoid aglycons [19,23], by utilizing analytical HPLC. Antiradical capacities of the extracts with DPPH and ORAC assays were also compared. In order to make comparison, the plants were harvested at the same flourishing period. To the best of our knowledge, this is the first report of these populations.

## 2. Results

### 2.1. Essential Oil Contents

As shown in Table 1, the studied plant samples were characterized mostly with similar EO yields. Populations “B” (1.03 ± 0.003%) and “BM” (0.78 ± 0.017%) contained the highest and lowest amount of EO, respectively (Table 1). As the plant sample “B” was cultivated in the university’s garden and it was regularly irrigated, the highest EO content can be predicted.

### 2.2. Chemical Profiles of Volatile Oils

Seventeen compounds were detected in these twelve populations. The sesquiterpene α-bisabolone oxide A (45.64–65.41%) was the major EO constituent in the samples except “B” and “S”, whereas its concentration was the highest in population “Mu” (Table 1).

The cultivated sample “B” was rich in α-bisabolol oxide B (21.88%) and chamazulene (19.22%), while the percentages of these compounds were lower under the wild growth conditions. The blue colour of EOs in all samples except “S” represented their chamazulene contents, whereas, “S” due to lack of this compound showed a yellowish green colour. Accordingly, oxygenated sesquiterpenes (53.31–74.52%) were the predominant chemical group of EO constituents in all studied samples, excluding “S”, which was rich in sesquiterpene hydrocarbons (Figure 1).

### 2.3. Apigenin and Luteolin Quantification of Methanolic Extracts

Methanolic extracts of samples “L” and “A” contained the highest amounts of apigenin, with 1.19 ± 0.01 mg/g and 1.02 ± 0.01 mg/g, respectively. Luteolin was present in higher concentrations in “BM” (2.20 ± 0.0 mg/g) and “A” (1.01 ± 0.02 mg/g) extracts (Figure 2).

### 2.4. Classification of M. Chamomilla Populations

To characterize and identify the different chemotypes of Iranian *M. chamomilla* populations, their EO compositions and main flavonoids (apigenin and luteolin) were submitted to cluster analysis (CA) and principal component analysis (PCA). As shown in Figure 3 and Figure 4, the dendrograms allowed separating the *M. chamomilla* populations into three main groups, each representing a distinct chemotype.

PCA is a mathematical procedure that transforms several correlated variables into various uncorrelated variables called principal components (PC). PC1, PC2, and PC3 showed the highest variation of phytochemicals among the studied populations. PC1 explained 41.57% of total variation and had a positive correlation with α-bisabolol oxide A, (*E*)-β-farnesene, α-bisabolone oxide A and (*E*)-spiroether, and negative correlation with (*Z*)-γ-bisabolene and (*E*)-γ-bisabolene. The second PC (PC2), with 24.98% of variance, demonstrated positive correlation with chamazulene, α-bisabolol oxide B, α-bisabolone oxide A and the EO content. Furthermore, PC3 represented positive correlation in the case of apigenin and luteolin, which accounted for 12.02% of the total variance (Table 2).

Regarding a significant contribution of phytochemical variation in PC1 and PC2, the scatter plot of PC1 and PC2 was used to specify phytochemical distance. The studied populations were classified in three groups, which confirmed the CA results.

In accordance with the CA, the populations “I”, “DS”, “Mo”, “MS”, “Mu”, “G”, “SS”, “L”, “A” and “BM” were classified into the same category, while, “B” and “S” were grouped into the individual subclasses. The first group possessed α-bisabolone oxide A and α-bisabolol oxide A as the major constituents as well as apigenin and luteolin (chemotype I). The second chemotype (II), was characterized by high amounts of chamazulene and α-bisabolol oxide B. The chemotype (III) was the richest in (*Z*) and (*E*)-γ-bisabolene.

### 2.5. Effect of Environmental Factors on Phytochemicals

In order to assess the effect of environmental factors on EO components, along with apigenin and luteolin contents, canonical correspondence analysis (CCA) was applied based on a matrix of three environmental factors including altitude, mean annual temperature (MAT), and mean annual precipitation (MAP) [24] and major EO compounds, along with apigenin and luteolin contents (Figure 5).

According to the CCA, phytochemicals of populations in first group were significantly affected by ecological parameters (altitude, temperature and precipitation) while the main oil composition and flavonoids of “Sarableh” and “Boldgold” were changed by genetic factors. Therefore, the “Sarableh” population can be introduced as a new chemotype.

### 2.6. Antiradical Activity of the Extracts

In the evaluation of the antioxidant potential of the twelve selected populations, “S” showed the most significant antiradical capacity, with EC_50_ = 7.76 ± 0.3 µg/mL and 6.51 ± 0.63 mmol TE/g measured by DPPH and ORAC assays, respectively. However, the extracts showed lower activity compared to ascorbic acid (EC_50_ = 0.3 ± 0.02 µg/mL) in the DPPH and rutin (20.22 ± 0.63 mmol TE/g) and EGCG (11.97 ± 0.02 mmol TE/g) in the ORAC assay (Figure 6 and Figure 7).

## 3. Discussion

The results of the present study confirm the variability of EO composition, flavonoid profile and antiradical activity, which are significantly affected by a diversity of ecological conditions.

The abiotic factors (e.g., moisture, topography, temperature, and edaphic factors) highly impact the phytoconstituents variation, and/or biotic factors remarkably influenced the terpene biosynthesis pathways and chemotype profiles [25]. However, the role of other ecological effects such as climatic factors (e.g., humidity, wind, atmospheric gases, and light), physiographic factors (e.g., steepness and sunlight on vegetation and direction of slopes), edaphic factors (the soil attributes), and biotic factors (e.g., the existence of lianas, epiphytes, parasites etc.) may also be considerable in chemotype variations.

In accordance with our findings, oxygenated sesquiterpenes (53.31–74.52%) are the most dominant EO compounds of the selected *Matricaria chamomilla* L. samples, which corroborates previous reports. The wild populations were significantly different compared with the cultivated sample (Bodgold) in EO composition. α-Bisabolone oxide A was mostly the major EO constituent in the populations.

In the literature, α-bisabolone oxide A was previously identified from *M. chamomilla* by different groups; for instance, the EO, diethyl ether and dichloromethane fractions of EO contained α-bisabolone oxide A, with 47.7, 57.7 and 50.5%, respectively [8]; whereas the EO of an Estonian *M. chamomilla* sample was characterized by high bisabolone oxide A content (13.9%) [2]. This compound was the major EO constituent of *M. chamomilla* grown in Italy (9.2–11.2%) [26], Iran (53.45%) [27], India (20.4 and 8.9%) [28] and Turkey (47.7%) [7]. The evaluation of the daily α-bisabolol oxide A content in *M. chamomilla* EO revealed that the highest amount can be between 16:00–18:00 (55.41%) [29]. Moreover, this aromatic phytoconstituent was isolated with three extraction methods (7.9, 42.3 and 50.5%) in EO of the species [9].

It is noteworthy, that the sample “Sarableh” (with the highest altitude and least minimum and maximum annual temperatures) as a new chemotype was the richest in two geometric isomers of γ-bisabolene (82.84%), whilst these sesquiterpenes were not markedly detected in the other plant populations. This population also demonstrated the most potent antiradical capacity compared with those samples.

γ-Bisabolene was previously specified as the major characteristic constituent of *Pimpinella pruatjan* Molk. [30]. EO of *Ocimum africanum* Lour. was also rich in (*E*)-γ-bisabolene (2.6–9.5%) [31]. This sesquiterpene was isolated from seeds of *Ziziphus jujuba* var. *inermis*. [32]. The main EO compounds of *M. chamomilla* harvested in Iran were identified as α-bisabolol (7.27%), (*Z*,*Z*)-farnesol (39.70–66.00%) [33], (*E*)-β-farnesene (24.19%) [34] and α-bisabolol oxide A (17.14%) [35]. EOs containing remarkable amounts of γ-bisabolene exhibited anti-inflammatory properties and anti-proliferative activities in human prostate cancer, glioblastoma, lung carcinoma, breast carcinoma, colon adenocarcinoma and human oral squamous cell lines [36,37,38,39].

According to the results, the abundance of luteolin was much higher than the apigenin content. The lack of luteolin and apigenin as the selected flavones was reported in populations “MS” and “Mo”. These samples are most probably rich in other phenolic natural products.

The total phenolic and flavonoid contents of a *M. chamomilla* extract were 37.51 and 21.72 mg/g of dry weight, respectively; this report was formerly accomplished by using HPLC-DAD [40]. Moreover, chlorogenic acid, apigenin-7-glucoside, rutin, cynaroside, luteolin, apigenin and apigenin-7-glucoside derivatives were previously qualified as the significant phytochemical composition of *M. chamomilla* extract [23].

As apigenin and luteolin were present in “Sarableh” in low concentrations, the high free radical scavenging capacity of this sample is obviously related to other polyphenolic compounds.

In a similar study, methanol extracts of *M. chamomilla* yielded from different samples indicated more potent antiradical activity than EOs analysed by the DPPH test, due to polyphenols present in extracts; although, they possessed a moderate effect in comparison with the standards [6].

Furthermore, in vitro antioxidant activities of *M. chamomilla* extracts, along with its apigenin and apigenin-7-glycoside contents were previously characterized, with IC_50_ of 18.19 ± 0.96 µg/mL, 2.0 ± 0.1 mg/g and 20.1 ± 0.9 mg/g, respectively [41], confirming the primary importance of apigenin in the antioxidant capacity of *M. chamomilla*. Moreover, free radical scavenging activity (DPPH assay) of *M. chamomilla* volatile oil and its major components were formerly recorded in the following order: chamazulene > α-bisabolol oxide A > chamomile EO > (*E*)-β-farnesene > α-bisabolol [7].

## 4. Materials and Methods

### 4.1. Plant Material

*M. chamomilla* flowers (300 g/sample) were individually collected from different regions of Iran in the flowering period, in spring (April) 2017. The harvested populations “Izeh”, “Bagh Malek”, “Lali”, “Masjed Soleyman”, “Mollasani”, “Gotvand” and “Saleh Shahr” from Khuzestan and “Murmuri”, “Abdanan”, “Darreh Shahr” and “Sarableh” from Ilam province were compared with “Bodgold”, which was cultivated at the botanical garden of Department of Horticultural Sciences, Shahid Chamran University of Ahvaz, Ahvaz, Iran.

The plants were identified by Dr. Mehrangiz Chehrazi affiliated with the same department, and a voucher specimen of each sample was deposited in the herbarium of the department. The voucher’s codes and geographic coordinates including the latitude, longitude, altitude using the Global Positioning System (GPS), along with mean annual temperatures of the studied populations are given in Table 3. The meteorological data (MAP, MAT, MMaxAT, and MMinAT) were collected from September until May 2017 [24]. The flowers were shade dried and finely ground. Each powdered sample was individually well-mixed and subjected to analysis.

### 4.2. Chemicals and Spectrophotometric Measurements

Analytical grade 1,1-diphenyl-2-picrylhydrazyl (DPPH), 2,2′-azobis-2-methylpropionamidine dihydrochloride (AAPH) and 6-hydroxy-2,5,7,8-tetramethylchroman-2-carboxylic acid (Trolox^®^) (Sigma-Aldrich, Steinheim, Germany), fluorescein (Fluka Analytical, Buchs, Germany); ascorbic acid, rutin and Na_2_SO_4_ (Merck, Darmstadt, Germany); epigallocatechin gallate (EGCG), apigenin (≥99%) and luteolin (≥97%) (Sigma-Aldrich, Germany) were purchased. Furthermore, all solvents of analytical grade were provided by the Merck company (Germany). Spectrophotometric measurements were carried out by a UV-VIS spectrophotometer (FLUOstar Optima, BMG Labtech, Ortenberg, Germany).

### 4.3. Extraction of Volatile Oils

To extract the EOs, 60 g of each powdered sample was individually subjected to Clevenger apparatus (hydro-distillation method) for 3 h. The obtained EOs were dried over anhydrous sodium sulphate and stored in refrigerator at 4 °C until analysis. Yields of extracted EOs were calculated by Equation (1):EO% = (EOs weight)/(dried plants weight) × 100(1)

Diethyl ether was used to elute the whole amount of EOs from the apparatus, and the weighing process was performed after evaporating the solvent.

### 4.4. Gas Chromatographic Analysis (GC-FID)

In the case of GC-FID analysis, the EOs were analyzed by Shimadzu GC-17A (Kyoto, Japan) equipped with an FID detector and SGE™ BP5 capillary column (Trajan Scientific and Medical, Victoria, Australia) (30 m × 0.25 mm column with a 0.25 μm film thickness). The split mode in GC was a ratio of 1:100. Injector and FID detector temperatures were set at 280 and 300 °C, respectively. The oven temperature was kept at 60 °C for 1 min and then raised to 250 °C at 5.0 °C/min and held for 2 min, while the ambient oven temperature range was +4 to +450 °C. Helium gas was used at a flow rate of 1 mL/min as a carrier gas.

### 4.5. Gas Chromatography-Mass Spectrometric (GC-MS) Analysis

Analysis of the samples was carried out using an Agilent 7890B gas chromatograph (Agilent Technologies, Inc., Santa Clara, CA, USA) equipped with a 5977B mass spectrometry detector. The GC instrument was equipped with a split inlet, working in a split ratio of 100:1 mode with a 30 m × 0.25 mm HP-5MS capillary column with a 0.25 μm film thickness and 0.25 μm particle size (temperature range: −60 to +320/340 °C). The injection port temperature was 250 °C. The oven temperature was kept at 60 °C for 1 min and then programmed from 60 °C to 250 °C at 5 °C/min; then, the temperature was kept at 250 °C for 2 min. Helium (99.999%) was used as a carrier gas with a flow rate of 1 mL/min and inlet pressure 35.3 kPa. The mass spectrometer was operated in the electron impact mode at 70 eV, and the inert ion source (HES EI) temperature was set to 350 °C; the temperature of the quadrupole was set at 150 °C, while the MS interface was set to 250 °C. A scan rate of 0.6 s (cycle time: 0.2 s) was applied, covering a mass range from 35 to 600 amu.

### 4.6. Identification of Essential Oil Components

Most of the compounds were identified using two different analytical approaches: (a) comparison of Kovats indices of *n*-alkanes (C_8_–C_24_) [42] and (b) comparison of mass spectra (using authentic chemicals and Wiley spectral library collection). Identification was considered tentative when based on mass spectral data alone. In GC-FID and GC-MS, data acquisition and analysis were performed using Chrom-card^TM^ (Scientific Analytical Solutions, Zurich, Switzerland, version DS) and Xcalibur^TM^ software (Thermo Fisher Scientific, Waltham, MA, USA, 4.0 Quick Start), respectively.

### 4.7. Preparation of Solvent Extracts

Five g of each sample was individually extracted with MeOH (3 × 75 mL) in an ultrasonic bath (VWR-USC300D) for 10 min, at 40 °C under power grade 9.

After removing the solvent under reduced pressure at 50 °C (Rotavapor R-114, Büchi), the concentrated extracts were subjected to evaluate the antiradical assays and HPLC analysis.

### 4.8. HPLC Analysis of Apigenin And Luteolin

Twenty μL of each extract (1 mg/mL) was separately injected into an analytical high-performance liquid chromatography system (HPLC) (Knauer, Berlin, Germany) by using an end capped Eurospher II 100-5 C18, Vertex Plus Column (Knauer, Berlin, Germany) (250 × 4.6 mm with precolumn) with particle size: 5 µm, pore size: 100 Å and temperature 30 °C; coupled to UV detector (Knauer GmbH-Smartline 2600, Berlin, Germany) at a wavelength range 190 to 500 nm (quantification at 330 nm), while MeOH/H_2_O was applied as the mobile phase with a gradient system, increasing MeOH from 30% to 70% within 40 min, with a flow rate of 1 mL/min, at ambient temperature. Analysis was performed using SAS software, version 9.2 (SAS Institute Inc., Cary, NC, USA).

### 4.9. Antiradical Capacity

#### 4.9.1. DPPH Assay

Free radical scavenging activity of the plant extracts was assessed by DPPH assay [43]. The measurement was carried out on 96-well microtiter plates. In brief, Microdilution series of samples (1 mg/mL, dissolved in MeOH) were prepared starting with 150 µL. To gain 200 µL of sample, 50 µL of DPPH reagent (100 µM) was added to each sample. The microplate was stored at room temperature under dark conditions. The absorbance was measured after 30 min at 550 nm using the microplate reader. MeOH (HPLC grade) and ascorbic acid (0.01 mg/mL) were used as a blank control and standard, respectively. Antiradical activity was calculated using the following Equation (2):I% = [(A_0_ − A_1_/A_0_) × 100](2)
where A_0_ is the absorbance of the control and A_1_ is the absorbance of the standard sample. Anti-radical activity of the samples was expressed as EC_50_ (concentration of the compounds that caused 50% inhibition). EC_50_ (µg/mL) values were calculated using GraphPad Prism software version 6.05 (GraphPad Software, San Diego, CA, USA). Each sample was measured in triplicate.

#### 4.9.2. ORAC Assay

Twelve extracts were subjected to ORAC assay [44] with slight modifications. Microtiter plates (96-well) were used for measurement of the samples. Briefly, 20 µL of extracts (0.01 mg/mL) was mixed with 60 µL of AAPH (peroxyl free radical generator) (12 mM) and 120 µL of fluorescein solution (70 mM). Then, the fluorescence was measured for 3 h at 1.5-min cycle intervals with the microplate reader. The standard Trolox^®^ was used. Activity of all samples was compared with rutin and EGCG as positive controls. Antioxidant capacities were reported as μmol TE (Trolox^®^ equivalents)/g of dry matter.

### 4.10. Statistical Analysis

All the experiments were done in triplicate, and the results are expressed as means ± SD. The data were assessed with one-way analysis of variance (ANOVA) using SAS software (version 9.2, SAS Institute Inc., Cary, NC, USA) and GraphPad Prism version 6.05. The means were compared using Duncan’s comparisons test (*p* < 0.05).

## 5. Conclusions

According to our preliminary study, the chemo-diversity and antiradical potential of twelve studied *Matricaria chamomilla* populations were highly affected by a variety of ecological conditions. To acquire the best yield with a good profile of active principles, it is crucial to combine a good genotype with optimal environmental circumstances.

In accordance with our findings, oxygenated sesquiterpenes are the most dominant EO compounds of the selected *Matricaria chamomilla* samples. Almost all plant populations contained apigenin and luteolin. Since the plant sample “Sarableh” indicated the most potent capacity to scavenge free radicals and its apigenin and luteolin contents were insignificant, its activity undoubtedly refers to the presence of other polyphenolic compounds. Due to high amounts of apigenin and luteolin, populations “Lali” and “Bagh Malek” can be considered as a rich source of these compounds. Our results confirm the effects of growth conditions on the quantity and quality of aromatic phytochemicals. More investigations are required to study the amounts of other polyphenolic compounds in diverse populations and the correlations between secondary metabolite contents, bioactivities and ecological effects.

## Figures and Tables

**Figure 1 molecules-24-01315-f001:**
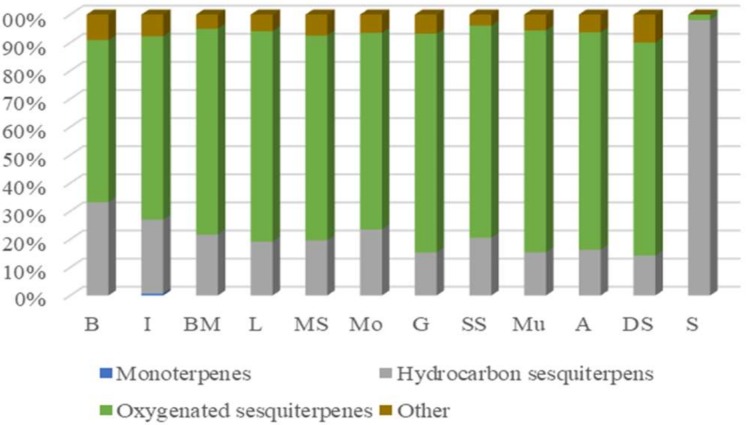
Volatile oil components from *Matricaria chamomilla* populations as a percentage of total identified compounds.

**Figure 2 molecules-24-01315-f002:**
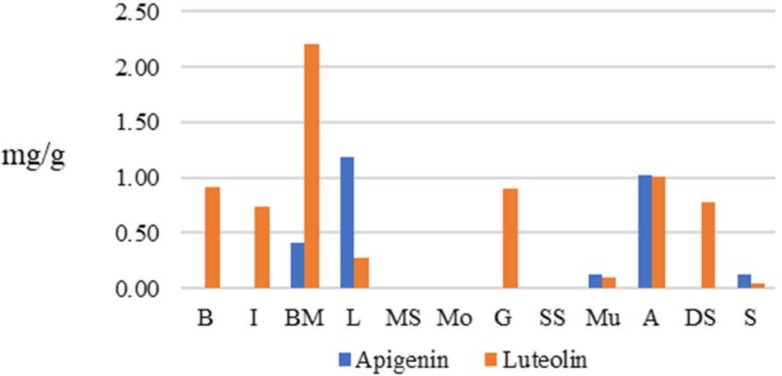
Apigenin and luteolin contents of twelve plant samples (mg/g of dry weight) analysed by HPLC.

**Figure 3 molecules-24-01315-f003:**
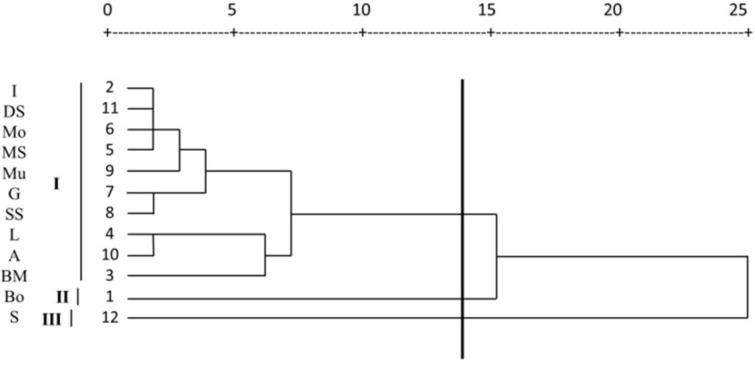
Dendrogram of the *Matricaria chamomilla* populations resulting from the cluster analysis (based on Euclidean distances) of the volatile oil components. chemotype I (α-bisabolone oxide A and α-bisabolol oxide A), chemotype II (chamazulene and α-bisabolol oxide B), chemotype III ((*Z*) and (*E*)-γ-bisabolene).

**Figure 4 molecules-24-01315-f004:**
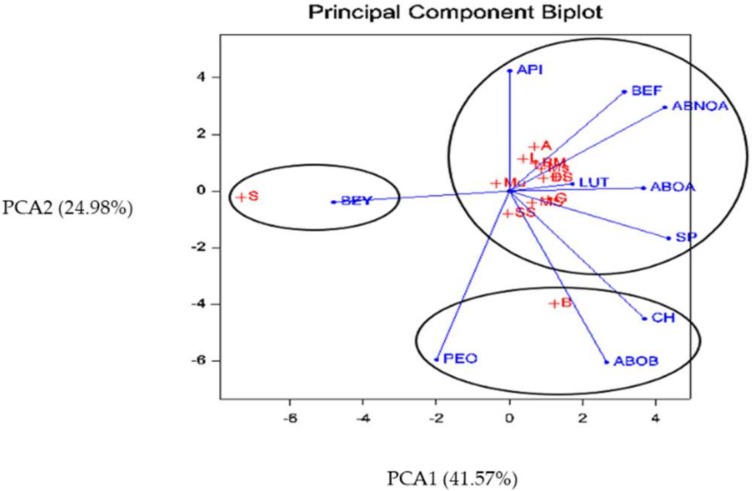
Principal component analysis (PCA) of the *Matricaria chamomilla* populations. ABOA: α-bisabolol oxide A, ABOB: α-bisabolol oxide B, PEO: percentage of essential oil, CH: chamazulene, SP: (*E*)-spiroether, ABNOA: α-bisabolone oxide A, BEF: (*E*)-β-farnesene, BZY: (*Z*)-γ-bisabolene, BEY: (*E*)-γ-bisabolene, LUT: luteolin, API: apigenin.

**Figure 5 molecules-24-01315-f005:**
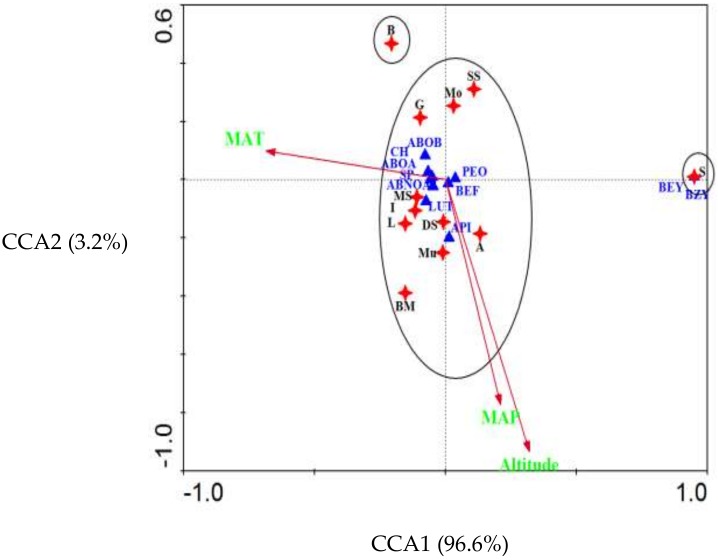
Canonical correspondence analysis (CCA) biplot of *Matricaria chamomilla* populations, linking percentages of the major constituents, collected from different environmental conditions. Populations B: Bodgold, I: Izeh, BM: Bagh Malek, L: Lali, MS: Masjed Soleyman, Mo: Mollasani, SS: Saleh Shahr, Mu: Murmuri, A: Abdanan, DS: Darreh Shahr, and S: Sarableh, MAT: mean annual temperature; MAP: mean annual precipitation; ABOA: α-bisabolol oxide A, ABOB: α-bisabolol oxide B, PEO: essential oil percentage, CH: chamazulene, SP: (*E*)-spiroether, ABNOA: α-bisabolone oxide A, BEF: (*E*)-β-farnesene, BZY: (*Z*)-γ-bisabolene, BEY: (*E*)-γ-bisabolene, LUT: luteolin, API: apigenin.

**Figure 6 molecules-24-01315-f006:**
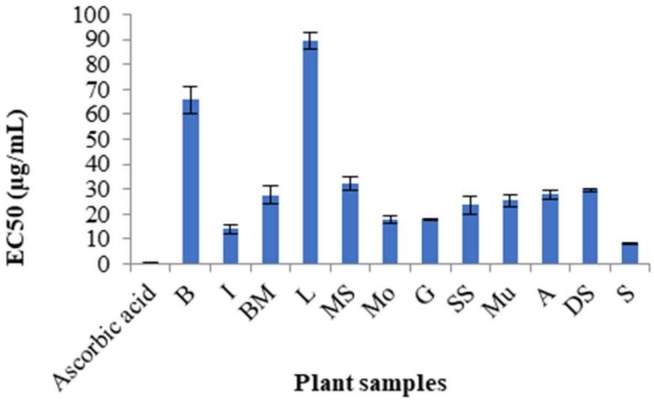
Antiradical scavenging activity of twelve plant samples of *Matricaria chamomilla* in the DPPH assay.

**Figure 7 molecules-24-01315-f007:**
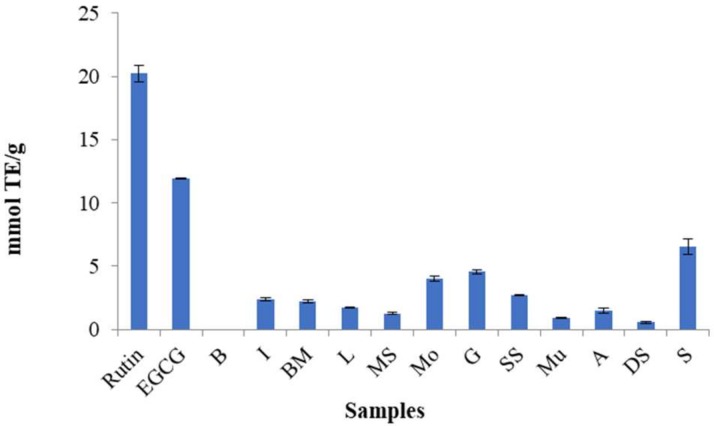
Antiradical capacity of selected *Matricaria chamomilla* populations in the ORAC assay.

**Table 1 molecules-24-01315-t001:** Essential oil constituents and yields of twelve harvested *Matricaria chamomilla* populations.

No.	RI ^A^	RT ^B^		Populations	B	I	BM	L	MS	Mo	G	SS	Mu	A	DS	S
			Compounds ^C^		Amount (%)											
1	1063	7.79	Artemisia ketone		nd	0.65	nd	nd	nd	nd	nd	nd	nd	nd	nd	nd
2	1423	17.75	*Trans*-caryophyllene		nd	nd	nd	nd	nd	nd	nd	nd	nd	nd	nd	0.5
3	1454	18.75	(*E*)-β-Farnesene		8.51 ^b^	16.68 ^a^	9.58 ^a,b^	10.60 ^a,b^	13.92 ^a,b^	13.91 ^a,b^	9.82 ^a,b^	6.61 ^b^	10.33 ^a,b^	15.29 ^a^	12.37 ^a,b^	5.25 ^b^
4	1484	19.45	Germacrene D		1.31	nd	nd	nd	nd	nd	nd	nd	nd	0.55	nd	2.21
5	1500	20	Bicyclo-germacrene		2.01	nd	nd	nd	nd	nd	nd	nd	nd	nd	nd	nd
6	1506	20.21	(*Z*)-α-Bisabolene		nd	nd	nd	nd	nd	0.81	nd	nd	nd	nd	nd	nd
7	1514	20.3	(*Z*)-γ-Bisabolene		nd	nd	nd	nd	nd	nd	nd	nd	nd	nd	nd	40.08
8	1529	20.7	(*E*)-γ-Bisabolene		nd	nd	nd	nd	nd	nd	nd	nd	nd	nd	nd	42.76
9	1561	21.65	(*E*)-Nerolidol		1.76	nd	nd	nd	nd	nd	nd	nd	nd	nd	nd	nd
10	1577	22.1	(+)-Spathulenol		1.53	nd	nd	nd	nd	nd	nd	nd	nd	nd	nd	nd
11	1630	23.18	(γ)-Eudesmol		nd	nd	nd	nd	nd	nd	nd	nd	nd	nd	nd	1.78
12	1656	24.16	α-Bisabolol oxide-B		21.88 ^a^	1.55 ^b,c^	1.52 ^b,c^	1.59 ^b,c^	1.39 ^c^	1.65 ^b,c^	1.68 ^b,c^	1.80 ^b^	1.58 ^b,c^	1.64 ^b,c^	1.37 ^c^	nd
13	1685	24.56	α-Bisabolol		nd	nd	nd	nd	1.32	2.86	2.07	nd	nd	nd	nd	nd
14	1693	24.9	α-Bisabolone oxide A		11.36 ^d^	47.91 ^b,c^	53.28 ^b^	52.14 ^b,c^	46.98 ^c^	46.74 ^c^	45.64 ^c^	51.87 ^b,c^	65.41 ^a^	47.7 ^b,c^	49.18 ^b,c^	nd
15	1730	25.76	Chamazulene		19.22 ^a^	8.29 ^b,c^	9.74 ^b^	4.74 ^e^	8.44 ^b,c^	6.14 ^d^	8.02 ^c^	9.3 ^b,c^	4.18 ^e^	2.58 ^f^	6.06 ^d^	nd
16	1748	26.18	α-Bisabolol oxide A		16.78 ^b,c^	14.03 ^c^	13.93 ^c^	19.35 ^b^	16.75 ^b,c^	19.41 ^b^	24.02 ^a^	22.26 ^a,b^	7.53 ^d^	20.25 ^b^	17.22 ^b,c^	nd
17	1890	26.31	(*E*)-Spiroether		8.37 ^a^	7.51 ^b^	4.70 ^c^	5.75 ^b,c^	7.12 ^a,b^	6.41 ^b^	6.49 ^b^	3.73 ^c^	5.31 ^b,c^	5.96 ^b^	7.26 ^a,b^	nd
**Total identified compounds %**					92.73	96.64	92.76	94.2	95.27	96.51	97.71	95.59	94.35	94.53	93.47	93.08
**EOs yield %**					1.03 ± 0.003	0.84 ± 0.006	0.78 ± 0.17	0.88 ± 0.012	0.94 ± 0.035	0.79 ± 0.006	0.88 ± 0.021	0.91 ± 0.009	0.9 ± 0.023	0.89 ± 0.006	0.83 ± 0.009	0.98 ± 0.021

^A^ Relative retention index to C_8_–C_24_
*n*-alkanes on HP-5MS column; ^B^ Retention times; ^C^ Compounds listed in order of elution from HP-5MS column; nd: not detected; the means were compared using Duncan’s comparisons test (*p* < 0.05); small letters (^a, b, c^) in each row show the significant difference of related component among various populations.

**Table 2 molecules-24-01315-t002:** Eigenvalues, variance and cumulative variance for three principal components.

Major Phytochemicals	Principal Components		
	PC1	PC2	PC3
Chamazulene	0.377	0.881	0.023
α-Bisabolol oxide A	0.760	0.126	0.114
(*E*)-β-Farnesene	0.679	−0.293	−0.004
α-Bisabolol oxide B	0.044	0.961	0.011
α-Bisabolone oxide A	0.742	0.525	0.104
(*E*)-Spiroether	0.849	0.377	−0.032
(*Z*)-γ-Bisabolene	−0.965	−0.119	−0.130
(*E*)-γ-Bisabolene	−0.965	−0.119	−0.130
Essential oil content	−0.480	0.600	−0.348
Apigenin	0.002	−0.295	0.638
Luteolin	0.160	0.235	0.857
Eigen values	4.57	2.74	1.32
Variance (%)	41.57	24.98	12.02
Cumulative variance (%)	41.57	66.55	78.57

**Table 3 molecules-24-01315-t003:** Geographic locations and climatic conditions of the studied *Matricaria chamomilla* populations from Iran.

Population Name	Voucher’s Code	Abbreviated Name	Altitude (m)	Latitude	Longitude	MAP (mm/year)	MAT (°C)	MMaxAT (°C)	MMinAT (°C)
Bodold (Ahvaz)	KHAU_236	B	16	31°18′ N	48°39′ E	98.20	22.62	35.77	9.47
Izeh	KHAU_237	I	428	31°57′ N	48°49′ E	472.80	18.35	31.28	5.42
Bagh Malek	KHAU_238	BM	907	31°19′ N	50°05′ E	285.90	20.28	32.17	8.4
Lali	KHAU_239	L	373	32°20′ N	49°05′ E	280.60	21.26	33.57	8.95
Masjed Soleyman	KHAU_240	MS	250	32°02′ N	49°11′ E	241.30	22.67	34.27	11.07
Mollasani	KHAU_241	Mo	51	31°39′ N	48°57′ E	100.80	22.26	36.35	8.17
Gotvand	KHAU_242	G	70	32°14′ N	48°48′ E	159.70	21.08	34.96	7.21
Saleh Shahr	KHAU_243	SS	65	32°04′ N	48°40′ E	164.30	20.69	34.07	7.32
Murmuri	KHAU_244	Mu	530	32°46′ N	47°37′ E	213.90	23.35	35.17	11.53
Abdanan	KHAU_245	A	740	32°55′ N	47°31′ E	363.60	19.62	30.44	8.8
Darreh Shahr	KHAU_246	DS	629	33°05′ N	47°28′ E	463.00	17.85	30.9	4.8
Sarableh	KHAU_247	S	1037	33°47′ N	46°35′ E	345.80	15.01	27.31	2.71

MAP: mean annual precipitation; MAT: mean annual temperature; MMaxAT: mean maximum anuual temperature; MMinAT: mean minimum anuual temperature.

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
