# Peer review of "Chemo-Diversity and Antiradical Potential of Twelve Matricaria chamomilla L. Populations from Iran: Proof of Ecological Effects"

_molecules, 2019, doi:10.3390/molecules24071315_

Round 1

Reviewer 1 Report

This work presents a consistent argumentation and all data are clear and described in details.

- Introduction: clear and argumentative.

- Results: PCA and HCA show clearly the importance of studies like that. 

This shows the engagement of the authors to understand effectively the Matricaria chamomilla L 

- Fig 3. Please justify more clearly why 3 groups and not 4 ?

- Fig 4. The explained variance is below 70%. Can you be more argumentative about that? 

- In terms of Gas chromatographic analysis (GC-FID), Gas chromatography-mass spectrometric (GC-MS) analysis and HPLC analysis of apigenin and luteolin:

- Were all these methods analytically validated?  

Author Response

answer attached

Reviewer 2 Report

The article entitled Chemo-diversity and antiradical potential of twelve Matricaria chamomilla L. accessions from Iran: a proof of ecological effects is an intersting one. It presents 12 different samples from 12 different places from Iran in order to achieve the influence of different factors on chemical constituents.

Please explain me why you use the noun Accessions it is the first time when I am finding into a scientific paper

The antiradical activity (DPPH and ORAC) was measured assuming like standard ascorbic acid or rutin, why the authors do not used the luteolin or apigenin which were the polyphenols measured in their samples?

In the PCA the authors do not used the results from DPPH and ORAC analysis, please explain

The PCA section should be improved.

Author Response

answer attached

Reviewer 3 Report

Comments

This paper describes the analyses of 17 volatile components, apigenin, and luteolin from 12 Matricaria chamomilla which harvested in various area in Iran. Authors claimed that the almost plants samples were featured by alpha-bisabolone oxide A and the other two were categorized different category. One of success is that the typical chemicals in M. chamomilla were revealed. On the other hand, this reviewer thinks that the samples, repeatability, and impact of results were limited for discuss the scientific new knowledge and this manuscript is not reach the acceptance level of this journal.

Questions

1.       The quantitative data of each compound from each sample was seemed to be using only one sample. (60 g x 1 for GC analyses, 5 g x 1 for HPLC) Compounds might be unevenly distributed among different stocks. Discussions using average of some samples per each area are needed.

2.       Apigenin and luteolin may be obtained by the results of hydrolyze from their glucosides. Therefore, analyses of their glucosides or total flavonoids are also needed.

3.       It is difficult to understand meaning of the DPPH and ORAC assay for this study. What knowledge was brought to this chemotaxonomic study by these assay results?

Author Response

aaass

Reviewer 4 Report

l.55

'glucoside and luteolin'

-> Glucoside of what?

l.58

'quantitively'

-> Correct typo.

l.60

'as the main its flavonoids'

-> Correct sentence order.

l.75

'C Compounds listed in order of elution from HP-5MS column'

-> RIs and RTs related to HP-1 are in good order so what is the reason to report the same order on HP-5?

-> Clarify sufficiently.

'D' is nice but overwhelming

-> If it is important, let it to be big, if not so much - use brevier/small font.

cap.4.1 (!)

What were the amounts of plants averaged before sampling 3 times per 60g for determination of EO content or 3 times per 5g for FLW content?

-> Supply. Usage of non-representative samples is a main problem of many analytical works.

l.289 + table 2

Source of meteorological data with period of data collection are missing.

-> Supply with reference.

btw / Meteorological data were not included to results/discussion/PCA. Why?

cap.4.2

Sigma is not from Hungary, Fluka not from Japan and so on.

-> Always report headquarters.

l.309 (!)

Were the amounts of essential oil weighed or measured volumetric? 60g yields 0.45-0.60 g of oil. Oil is viscous and adhere to glass parts of apparatus. How to collect all and weight without significant loss?

-> Clarify or correct.

cap. 4.4-4.5 (!)

Calibration mixture (+ source) for RIs not reported. RIs cannot be transferred from HP-1 to HP-5. If HP-1 was not used for GC why to report RI's from HP-1 (from literature?)?

-> Supply and clarify.

l.339

Sonication parameters missing.

-> Supply.

cap.4.8, l.388-389 (!)

How was the amount of TOTAL api and lut calculated (Were glycosides recalculated to aglycones?)? What was the analytical wavelength? What was the purity of standards (with method)? ...

-> Supply and clarify.

Author Response

answer attached

Round 2

Reviewer 2 Report

Accept as it is

Author Response

Thank you for your positive opinion!

Reviewer 3 Report

Comments

This reviewer re-reviewed this manuscript, and understood that the manuscript was updated and improved.

On the other hand, it was not solved my questions 1. few trial times of analyses for each collected plant sample from each area (I didn’t ask about repeat times of HPLC and GC injections.); 2. The inadequacy of quantitative estimating only aglycone of flavonoids.

Above questions were in my opinion. I have been checked there are no scientific errors in the manuscript except for concerning above as far as my knowledge. Judgment of acceptance of this manuscript to the journal is left to the editor.

Stylistic

Line 68: Is Table 2 correct? I expected this is Table 3.

Line 101: Please start a new line.

Line 339: “+4 to +450 ℃” in accordance with line 346 is better than “+4-450 ℃”.

Table 2: Configurations of compound name (E) and (Z) should be italic style.

Figure 7: Go => G?

Author Response

Thank you for reviewing our manuscript. Please find our answers below.

The authors

Reviewer 3

This reviewer re-reviewed this manuscript and understood that the manuscript was updated and improved.

On the other hand, it was not solved my questions 1. few trial times of analyses for each collected plant sample from each area (I didn’t ask about repeat times of HPLC and GC injections.); 2. The inadequacy of quantitative estimating only aglycone of flavonoids.

1. From each population, around 1.5 kg of fresh flowers were harvested. After drying, 300 g of each plant material was finely powdered and mixed, individually.The mixed samples which were representative of that specific population were subjected to the phytochemical analysis and biological assays. The big amount of samples and the careful homogenization before and after grinding ensures that the gained results are representative of the analysed population. Unfortunately we are not able to do additional experiments with these samples.   

2. Our study aimed at the quantification of the major flavonoid aglycons apigenin and luteolin (as the major flavonoids in Matricaria chamomilla). The analysis of glycosides would need a different analytical approach and was out of the scope of our work.

Above questions were in my opinion. I have been checked there are no scientific errors in the manuscript except for concerning above as far as my knowledge. Judgment of acceptance of this manuscript to the journal is left to the editor.

Stylistic

Line 68: Is Table 2 correct? I expected this is Table 3.

- corrected.

Line 101: Please start a new line.

- done.

Line 339: “+4 to +450 ” in accordance with line 346 is better than “+4-450 ”.

- revised.

Table 2: Configurations of compound name (E) and (Z) should be italic style.

- corrected.

Figure 7: Go => G?

- done.